# Direct evidence of hidden local spin polarization in a centrosymmetric superconductor $LaO_{0.55}F_{0.45}BiS_2$

Shi-Long Wu[1], Kazuki Sumida [1], Koji Miyamoto[2], Kazuaki Taguchi[1], Tomoki Yoshikawa[1], Akio Kimura[1], Yoshifumi Ueda [2], Masashi Arita[2], Masanori Nagao[3], Satoshi Watauchi[3], Isao Tanaka[3] & Taichi Okuda [2]

Conventional Rashba spin polarization is caused by the combination of strong spin–orbit interaction and spatial inversion asymmetry. However, Rashba–Dresselhaus-type spin-split states are predicted in the centrosymmetric $LaOBiS_2$ system by recent theory, which stem from the local inversion asymmetry of active $BiS_2$ layer. By performing high-resolution spin- and angle-resolved photoemission spectroscopy, we have investigated the electronic band structure and spin texture of superconductor $LaO_{0.55}F_{0.45}BiS_2$. Here we present direct spectroscopic evidence for the local spin polarization of both the valence band and the conduction band. In particular, the coexistence of Rashba-like and Dresselhaus-like spin textures has been observed in the conduction band. The finding is of key importance for fabrication of proposed dual-gated spin-field effect transistor. Moreover, the spin-split band leads to a spin–momentum locking Fermi surface from which superconductivity emerges. Our demonstration not only expands the scope of spintronic materials but also enhances the understanding of spin–orbit interaction-related superconductivity.

[1] Graduate School of Science, Hiroshima University, 1-3-1 Kagamiyama, Higashi-Hiroshima 739-8526, Japan. [2] Hiroshima Synchrotron Radiation Center (HSRC), Hiroshima University, 2-313 Kagamiyama, Higashi-Hiroshima 739-0046, Japan. [3] Center for Crystal Science and Technology (CCST), University of Yamanashi, 7-32 Miyamae, Kofu, Yamanashi 400-8511, Japan. Correspondence and requests for materials should be addressed to T.O. (email: okudat@hiroshima-u.ac.jp)

According to the well-known Kramers theorem, spatial inversion symmetry $E(\mathbf{k},\uparrow) = E(-\mathbf{k},\uparrow)$ and time-reversal symmetry $E(\mathbf{k},\uparrow) = E(-\mathbf{k},\downarrow)$ result in $E(\mathbf{k},\uparrow) = E(\mathbf{k},\downarrow)$, guarantee that the electronic states of non-magnetic centrosymmetric materials must be spin degenerate. Namely, a momentum-dependent spin-split state usually comes from global inversion asymmetry of system. For instance, conventional Dresselhaus effect (denoted as D-1 in ref. [1]) can derive from non-polar bulk crystal with inversion asymmetry[2]. Alternatively, polar bulk materials[3] or surface or interface under electrostatic potential gradient leads to conventional Rashba effect (denoted as R-1)[4–7]. However, recent theoretical formalism reconstructed the traditional Rashba and Dresselhaus effects in atomic scale[1,8]. Microscopically, atomic site that belongs to a non-cetrosmmetric point group can carry either a local dipole field or site inversion asymmetric crystal field, inducing local Rashba effect (denoted as R-2) or local Dresselhaus effect (denoted as D-2), respectively[1,8].

The theoretical works suggested that LaOBiS2 and the related compounds can be such systems possessing R-2 and/or D-2 due to the breaking of local inversion symmetry in each BiS2 bilayer and the opposite polar fields caused by ionic bonding between $(BiS_2)^-$ bilayer and $(La_2O_2)^{2+}$ layer. Since the projected local spin polarization on each real-space sector of BiS2 bilayer in LaOBiS2 crystals holds opposite orientation, so called spin-layer locking effect, has been theoretically predicted in the LaOBiS2 film at first[9] which could offer advantages for the design of new generation of spin-field effect transistors (SFET)[9]. Theoretical study of either film[9] or bulk LaOBiS2[1] further pointed out that spin texture of conduction band at each X point in Brillouin zone must be non-helical originating from D-2 effect whereas the valence band possesses helical spin texture originating from R-2 effect.

Moreover, with electron doping by substitution of oxygen with fluorine, $LaO_{1-x}F_xBiS_2$ is manifested as one of BiS2-based superconductors with similar properties of cuprate superconductors such as rather high value of $2\Delta/k_BT_C$[10] and Cooper pairing symmetry[11]. The system exhibits the maximum superconducting critical temperature ($T_c$) of 10.6 K at $x \sim 0.5$[12] among all the BiS(Se)2-based superconductors, hence serves an excellent platform to combine superconductivity with local Rashba effect, by which we focus on a new approach to study the mechanism of Cooper pairing. Note that, heretofore, Rashba superconductors[13–16] with mixed singlet and triplet pairings have been limited to non-centrosymmetric compounds or surface systems.

Intriguingly, recent theoretical papers proposed that BiS2-based superconductors could be classified as time-reversal invariant (TRI) weak topological superconductor (TSc) as a result of combination of possible $d*_{x^2-y^2}$ pairing symmetry and possible spin-split electronic band at X points[17,18]. Therefore BiS2-based superconductors may have a dominant triplet pairing component arising from the spin–orbit interaction (SOI) and respecting time-reversal symmetry. Consequently the dominant triplet gap can cause gap sign changes between the spin-split Fermi pockets[17,18] which give rise to weak topological superconductivity[19]. The evidence of spin-polarized states caused by SOI and breaking of local inversion symmetry on LaOBiS2 or family compounds (LaOBiSe2, etc.), however, has not yet been reported so far.

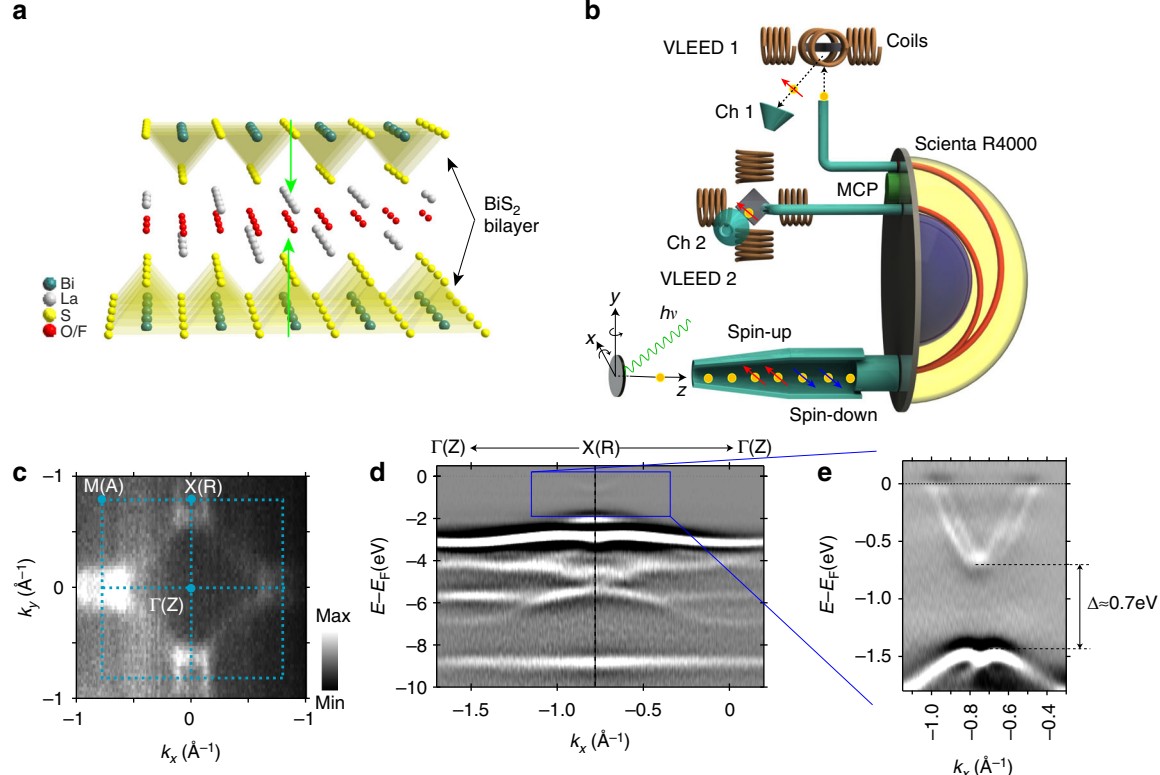

**Fig. 1** Electronic structure of LaO0.55F0.45BiS2 observed by angle-resolved photoemission (ARPES). **a** Crystal structure of La(O,F)BiS2 with a buffer layer La (O,F) separating the active BiS2 bilayer. Green arrows denote opposite local polar fields in BiS2 bilayer. **b** Illustration of Efficient SPin REsolved SpectroScOpy (ESPRESSO) machine at HiSOR and the experimental geometries for the ARPES and spin-ARPES measurements. **c** Constant energy contour (CEC) at $E_B = 0.2$ eV. **d** Band structure ($E-k$ map) along Γ(Z)-X(R) obtained by ARPES measurement taken with $h\nu = 70$ eV and $T_S = 50$ K. The CEC map is integrated over a window of ±20 meV and the $E-k$ map is obtained by the second derivative of energy distribution curves (EDCs) of original ARPES data. **e** Bands in the range of 0–1.8 eV below Fermi level (the boxed area of **d**) are clearly seen in the ARPES data taken with $h\nu = 18$ eV and $T_S = 50$ K

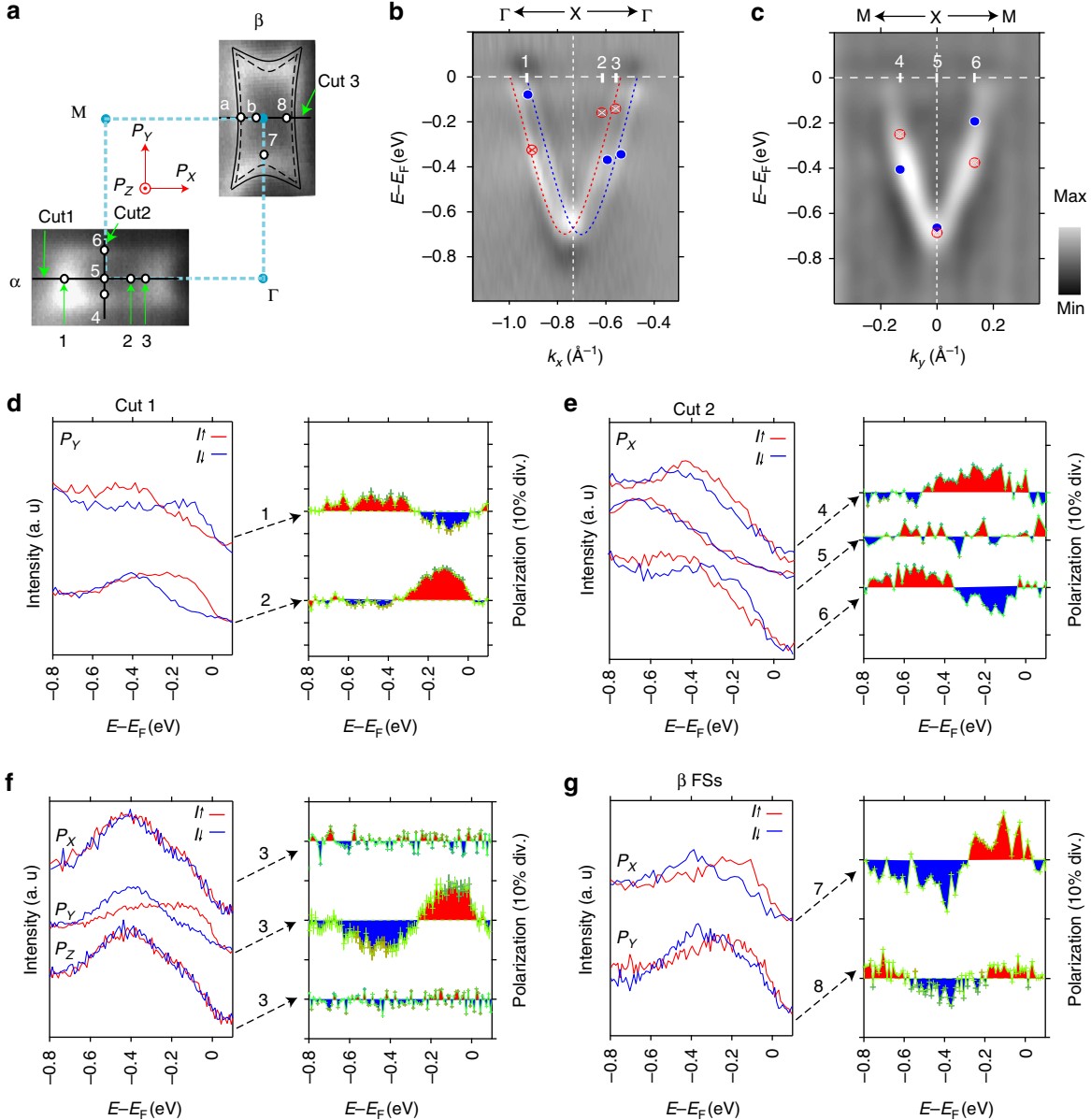

**Fig. 2** Spin- and angle-resolved photoemission spectroscopy (SARPES) of lowest conduction band (LCB). **a** Fermi surface sheets (FSs) labeled α and β of LCB measured by using photon energy of 18 eV and the comparison with FSs of DFT calculation (black lines). The constant energy contour (CEC) maps are integrated over a window of ± 20 meV. The dots around X point denote the momentum points where we performed spin measurements, the points 1–3 along cut 1 and the points 4–6 along cut 2 were also marked in **b**, **c**. Coordinate axes ($P_X$, $P_Y$, $P_Z$) denote positive directions of spin vectors. **b** Band dispersion measured by ARPES ($h\nu = 18$ eV) along the cut 1 (Γ–X–Γ line, second derivative). The dashed red and blue lines represent extracted positions from the energy distribution curves (EDCs) used for the estimation of Rashba parameter. **c** The same as **b** but along the cut 2 (M–X–M line, second derivative). **d** Spin-resolved EDCs of $P_Y$ and its spin polarization along cut 1. **e** The same as **d** but of $P_X$ along cut 2. The peak positions of spin-up and spin-down states along cut 1 and cut 2 are indicated in **b** and **c** by red crosses and blue dots. **f** The same as **d** but of $P_X$, $P_Y$ and $P_Z$ at point 3. **g** The same as **d** but of $P_X$ at point 7 and $P_Y$ at point 8

In the following, we provide the direct evidence for the hidden local spin polarization in the vicinity of time-reversal invariant momenta (TRIM) X points of both conduction band (CB) and valence band (VB) in $LaO_{0.55}F_{0.45}BiS_2$ superconductor by high-resolution spin- and angle-resolved photoemission spectroscopy (SARPES). Especially the conversion from Rashba-like to Dresselhaus-like spin texture with varying binding energy has been directly observed in the conduction band. Our observation of the unconventional spin-split state in $LaO_{0.55}F_{0.45}BiS_2$ not only promotes $BiS_2$-based materials as an important platform for realizing spintronic devices but also enhances the understanding of SOI-related superconductivity.

## Results

**Electronic states of $LaO_{0.55}F_{0.45}BiS_2$.** Figure 1a shows the crystal structure of $LaO_{1-x}F_xBiS_2$ possessing a P4/nmm symmetry with an inversion center located at the center of two nonequivalent O atoms, stacked alternatively an active $BiS_2$ bilayer and a buffer La (O,F) layer[12,20–22]. The green arrows in Fig. 1a show the dipole fields along normal direction of BiS plane induced by ionic bonding between $(BiS_2)^-$ bilayer and $(La_2O_2)^{2+}$ layer. The sandwich structure that consists of the $BiS_2$ and the La(O,F) layer stacks with weak van der Waals force. Therefore, the sample can be cleaved at the van der Waals gap and the surface is always terminated by $BiS_2$ layer.

Figure 1c shows the constant energy contour (CEC) map of LaO$_{0.55}$F$_{0.45}$BiS$_2$ grown by flux method[23] at binding energy ($E_B$) of 0.2 eV taken with ESPRESSO machine[24,25] (Fig. 1b) (see Methods section below for more detailed information). Rectangle-like Fermi pocket locates around each X symmetry point in the Brillouin zone, reflecting the four-fold symmetry of crystals. Figure 1d shows the band structure along Γ(Z)–X(R) (Γ–X for short) high-symmetry direction obtained by the second derivative of energy distribution curves (EDCs) taken at $hν = 70$ eV. (See Supplementary Fig. 1 for the raw data.) In accordance with the previous density functional theory (DFT) calculations[26,27], several states from 2 to 6 eV below Fermi level ($E_F$) are observed in valence band, which are attributed mainly to the 2$p$ states of O and S atoms. Although the intensity of highest valence band (HVB) and lowest conduction band (LCB), which shift below $E_F$ by fluorine doping, is relatively weak in the boxed area of Fig. 1d, it is quite clear in Fig. 1e taken with $hν = 18$ eV after the second derivative processing.(See also raw data in Supplementary Fig. 1) The splitting electron-like conduction band crosses $E_F$ around X point coming from Bi-6$p$ state[26,27], so that the BiS plane dominantly contributes to the electronic conduction. Figure 1d and e also show that, along the Γ–X direction, the maximum value of valence band is at the X point with a bandgap of about 0.7 eV to the bottom of the conduction band being consistent with the band calculation[27]. We note the overall features of electronic structure in Fig. 1 are in good accordance with the previous calculations and experiments[1,8,27] for the bulk electronic states in BiS$_2$-based compounds.

**Spin-split LCB**. In Fig. 2, we present more detailed band structure obtained by ARPES measurement with $hν = 18$ eV around X symmetry point. Figure 2a shows CEC images at the Fermi level (i.e., Fermi surface sheets (FSs)) and the calculated FSs for LaO$_{1−x}$F$_x$BiS$_2$ ($x ~ 0.45$) (black lines) plotted from previous studies[11,27] for comparison. The rectangle-like shape and its size are corresponding to doping level of ~0.45 in LaO$_{1−x}$F$_x$BiS$_2$ system. Figure 2b shows the splitting LCB along Γ–X–Γ line (cut 1 in Fig. 2a). In contrast to Fig. 2b, the LCB along M–X–M line in Fig. 2c (cut 2 in Fig. 2a) does not show clear band splitting which is consistent with the expected smaller splitting along the direction by the DFT calculation[1].

Given that pure Rashba-type spin polarized HVB as well as the LCB in LaOBiS$_2$ system, namely assuming that the band splitting from Fig. 1e (Fig. 2b) is due to Rashba spin–orbit interaction (RSOI), we can estimate that the Rashba energies $E_R = \hbar^2 k_R^2/2 m^\star$ ($k_R$ describes the shift between band extremum and crossing, $m^\star$ denotes the effective mass) are 24 meV for LCB and 45 meV for HVB. Since the corresponding momentum offsets $k_R = \alpha_R m^\star/\hbar^2$ are 0.04 Å$^{-1}$ and 0.08 Å$^{-1}$, we can evaluate the Rashba parameters $\alpha_R = 2E_R/k_R = 1.20$ and 1.13 eV Å for LCB and HVB, respectively. The magnitudes of the Rashba parameter for LCB is in fair agreement with the predicted value 1.24 eV Å while that for HVB is a little smaller than the predicted value 1.84 eV Å for HVB[8].

In order to investigate the real origin of the splitting in LCB and HVB we have performed SARPES measurement utilizing the function of three-dimensional spin-vector analysis of ESPRESSO machine[25] at the representative momentum points marked by white dots in Fig. 2a. Figure 2d–g show the SARPES results of LCB at several momentum points around X with 3D spin-vector analysis. Figure 2d illustrates the spin resolved energy distribution curves (spin-EDCs) of $P_Y$ (=tangential spin to the Fermi surface) at the momentum positions 1 and 2 along Γ–X–Γ line (cut 1). In the figure, the red (blue) lines denote spin-up (spin-down) states. The definition of positive directions of spin vectors are shown with the coordinate axes in Fig. 2a. The results indicate that the

LCB is unambiguously spin polarized and the spin-splitting occurs at each measuring momentum position. Moreover, the spin reversal can be observed on opposite sides of X point. Namely, the sign of spin polarizations at position 1 is opposite to that of position 2 in the spin-resolved EDC spectra.

In contrast with the obvious spin polarization of $P_Y$, spin polarizations of $P_X$ and $P_Z$ are negligible at position 3 as shown in Fig. 2f indicating the spin polarization is mainly along in-plane tangential direction of Fermi surface. To further clarify the spin texture of the whole FSs, we have also performed spin measurement of $P_X$ (= tangential spin to the Fermi surface) at positions 4, 5, and 6 along M–X–M line (cut 2) by tilt rotation of sample. The almost identical spectra in opposite spin channels of $P_X$ at position 5 (i.e., at the X(R) high-symmetry point) confirm the spin degeneracy of LCB at the time-reversal invariant momenta (TRIM). Although the band splitting is not obvious in normal ARPES measurement in Fig. 2c, clear spin polarizations and its reversal with respect to X symmetry point are also confirmed in the M–X–M line as shown in the Fig. 2e. The peak positions of spin-resolved EDCs in Fig. 2d, e are plotted in Fig. 2b, c with several marks (i.e., dots and crosses) being in reasonable agreement with the observed band dispersions by normal ARPES measurement. The observed spin reversals of $P_X$ along M–X–M line and $P_Y$ along Γ–X–Γ line indicate the overall counter-helical spin texture for LCB, which strongly suggests that Rashba spin polarization occurs in LCB.

On the other hand, it is also known that various experimental geometries such as changes of incident angle of synchrotron-radiation (SR) light, photon energy and so on could affect the observed spin polarization especially in the system with strong SOI[28]. Thus, to make a double check of our results, we have performed similar measurement in β slice of FSs. In addition, with comparison of measurements of α and β slices we can not only confirm the Rashba-type spin polarization but also investigate the spin texture of FSs in the whole Brillouin zone. Figure 2g shows the spin-resolved EDC spectra and their spin polarizations at positions 7 and 8 in β FSs. Clear spin polarizations and its sign of $P_X$ at position 7 and $P_Y$ at position 8 again confirm the helical spin texture around X symmetry point in β FSs. In other words, the directions of polarizations at positions 7 and 8 are equivalent to those of positions 4 and 3, respectively. As a result, the LCB around each high symmetry X point in Brillouin zone has the same counter-helical spin texture.

Therefore, SARPES measurement confirmed that each slice of rectangle-like FSs has a counter-helical spin texture as summarized in Fig. 3c. Furthermore, the Rashba-like counter-helical spin texture, being consistent with the theoretical prediction, is also strongly suggested in HVB by our SARPES measurement (Supplementary Fig. 2).

However, we must confirm the bulk contribution in the observed rectangle-like shape FSs so as to conclude that the counter-helical spin texture is attributed to local Rashba effect (R-2 effect) instead of surface Rashba spin splitting (R-1 effect). Recently bulk sensitive soft X-ray ARPES using photon energy of 880 eV was performed with LaO$_{0.54}$F$_{0.46}$BiS$_2$ crystal[27], which concluded that the observed rectangle-shape FSs by VUV-ARPES with $hν = 70$ eV is the contribution of bulk state because of the close similarity of FSs between $hν = 880$ and 70 eV measurement which is almost identical to the one obtained with $hν = 18$ eV in our experiment. In addition, the overall agreement between calculated bulk band structures and experimentally observed band dispersions also strongly suggests that the observed LCB band is bulk state of LaO$_{0.55}$F$_{0.45}$BiS$_2$ crystal. Therefore, we believe that the observed spin polarization at LCB is attributed to local Rashba effect (R-2).

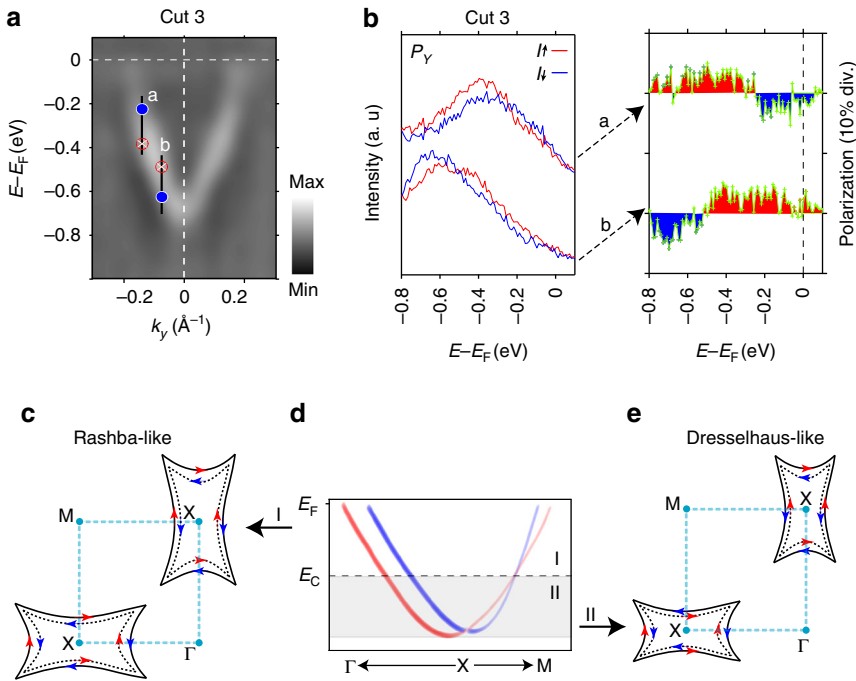

**Fig. 3** Summary of the observed transition from Rashba-like to Dresselhaus-like spin textures for LCB. **a** Band dispersion measured by ARPES ($h\nu = 18$ eV) along the cut 3 in Fig. 2a (X–M line). **b** Spin-resolved EDCs of $P_Y$ at a and b points in Fig. 2a, right panel shows the corresponding spin polarizations. The peak positions of spin-resolved EDCs are indicated with crosses and dots in **a**. The reversal of spin polarizations at a and b points leading to a different spin textures (Rashba-like and Dresselhaus-like) in LCB. **c** The Rashba-like spin texture of the rectangle-like shape FSs at upper part of LCB (region I in **d**). **e** The Dresselhaus-like spin texture of CECs at lower part of LCB (region II in **d**) derives from the crossing of spin-polarized LCB at $E = E_C$ along X–M line, which was taken from refs. [1,8] and schematically indicated in **d**. The red and blue colors for arrows and energy bands represent spin-up and spin-down states, respectively

At a first glance, nevertheless, our observation of Rashba-type spin texture in LCB is being inconsistent with the theoretical prediction in which the Dresselhaus-type spin texture was expected for the LCB[1]. However, as indicated in Fig. 3d, in the theory paper, band crossing is predicted along X–M line which divides LCB into upper and lower parts (regions I and II) in LaOBiS$_2$ compound[1,8,9] such that the lower LCB has non-helical spin texture whereas the upper LCB possesses helical spin texture suggesting the coexistence of D-2 and R-2 spin polarizations (Supplementary Figs. 3 and 4). In order to investigate the novel phenomenon we have performed further SARPES observation very near to the X point along X–M line as in Fig. 3a, b. Intriguingly, we found that the spin polarization close to the X point is opposite to that of momentum position further from the X point strongly suggesting the transition from Dresselhaus-like to Rashba-like spin texture with varying binding energy.(see Supplementary Fig. 5 for the entire SARPES data along X–M line.) Therefore our observation is in good agreement with previous theoretical prediction though the crossing point seems to be a little nearer to the X point in the experiment than theoretical prediction. Furthermore, the observed transition from Dresslhaus-like to Rashba-like spin texture also strongly supports the idea that the observed Rashba-type spin texture is mainly due to bulk contribution (i.e., R-2 effect) as discussed above since the normal Rashba effect by electrostatic potential gradient (R-1) does not cause the Dresselhaus-type spin texture.

## Discussion

The reason why we can extract the R-2 spin polarization by photoemission is probably due to the short mean free path of the photoemission measurement using proper energy photon as demonstrated in transition metal dichalcogenides (TMDCs)

recently[29–31]. In addition, unlike the TMDCs, in case of LaOBiS$_2$ and its family compounds there is no possibility of existence of different domains having opposite spin polarization because the sample is cleaved at van der Waals gap between adjacent BiS$_2$ bilayers and surface is always terminated by BiS$_2$ layer. Therefore even with relatively large beam spot size one can fairly observe the spin polarization.

From an application point of view, our finding of spin-active BiS$_2$ bilayer opens the pathway to realize the LaOBiS$_2$-based fast SFET. Since the opposite spin polarizations lock with BiS$_2$ bilayer, it is much easier to select a different layer so as to obtain reversal spin-polarized state via applying a small electric field[9]. Further exploration of local spin-polarization on iso-structural materials such as LaOBiSe$_2$[32], (LaO)$_2$(SbSe$_2$)$_2$[33] and so on will play an essential role in fabrication of spintronic devices.

Finally, we would briefly discuss the relationship between the local Rashba spin polarization and superconductivity. R-2 spin polarization corresponds to upper LCB near the Fermi level as shown in Fig. 3c such that the spin-momentum locking Fermi surface could have profound effects on the emergence of super-conductivity in La(O,F)BiS$_2$. There are a broaden range of lit-eratures to address unconventional superconductivity with inversion asymmetry in non-centrosymmetric heavy fermion compounds[13–16]. Because the broken inversion symmetry induces RSOI, as a result, different parities, spin-singlet and spin-triplet pairing, can be mixed in a superconducting state. Our experi-mental results evidently demonstrate the superconducting BiS$_2$ layers are also spin-active, thereby singlet and triplet pairings can be mixed in the wave function of the Cooper pairs[13–15]. These observations may suggest a new approach to enhance super-conducting critical temperature ($T_c$) by increasing strength of RSOI in the BiS$_2$-based system[16]. Furthermore, it is suggested that

the spin-active layers can have nontrivial topology if the super-conducting gap wavefunction has opposite signs on the spin-momentum locking Fermi surface[17–19]. The local spin polarization of electron pockets directly observed in present study acts as an essential ingredient, could lead to intrinsic topological super-conductivity in the $BiS_2$-based system.

In conclusion, we have found that SOI plays a significant role in the electronic energy bands of centrosymmetric super-conductor $LaO_{0.55}F_{0.45}BiS_2$ with globally centrosymmetric crystal structure. Using high-resolution spin- and angle-resolved pho-toemission spectroscopy, we directly observed the hidden local Rashba and Dresselhaus spin polarization in a centrosymmetric superconductor. Unlike the hidden Rashba state observed recently in transition metal dichalcogenides transition from Dresselhaus-like to Rashba-like spin texture is revealed in this material. Since the spin-split Fermi pockets demonstrated by present experiment could have significant effects on Cooper paring and topological properties, we expect further works in $BiS_2$-based systems can promote studies of topological super-conductors (TSc) and Marjorana fermions. Therefore, not only the result is of key importance for applying these compounds to SFET, but also offers an accessible paradigm to probe, manipulate bulk spin polarization based on local asymmetry and search TSc on much wider area of bulk materials.

## Methods

**Crystals growth**. La(O,F)$BiS_2$ single crystals were grown by a high-temperature flux method in a vacuumed quartz tube. The raw materials of $La_2S_3$, Bi, $Bi_2S_3$, $Bi_2O_3$, $BiF_3$ were weighed to have a nominal composition of $LaO_{0.4}F_{0.6}BiS_2$. A mixture of raw materials (0.8 g) and CsCl/KCl flux (5.0 g) with a molar ratio of 5:3 was combined using a mortar, and then sealed in a quartz tube under vacuum. The quartz tube was heated at 800 °C for 10 h, cooled slowly to 600 °C at a rate of 1 °C h$^{-1}$, and then furnace-cooled to room temperature. The quartz tube was opened in air, and the flux was dissolved in the quartz tube using distilled water. $LaO_{0.55}F_{0.45}BiS_2$ single crystals were obtained in this product. The obtained single crystals had good cleavage, producing flat surfaces as large as ~$1 \times 1$ mm$^2$.

**(S)ARPES experiments**. Both ARPES and SARPES measurements were per-formed at ESPRESSO endstation[25] of Hiroshima Synchrotron Radiation Center (HiSOR). The VLEED-type spin polarimeter utilized in the ESPRESSO achieves a 100 times higher efficiency compared to that of conventional Mott-type spin detectors, which offers a great advantage for high-resolution spin analysis of nonmagnetic system as in present case. The spin polarizations of photoelectrons can be measured by switching the directions of the target magnetizations by coils such that ESPRESSO machine can resolve both out-of-plane ($P_Z$) and in-plane ($P_X$/$P_Y$) spin polarization components with high angular and energy resolutions[26]. The sign of the polar (tilt) angle is defined as negative (positive) in the case of antic-lockwise rotation about $y$ axis ($x$ axis) as shown in Fig. 1b. The overall experimental energy and wave number resolutions of ARPES (SARPES) were set to 35 meV and <0.008 Å$^{-1}$ (60 meV and <0.04 Å$^{-1}$), respectively. The samples were cleaved in-situ under ultrahigh vacuum(UHV) below $1\times10^{-8}$ Pa and the sample temperature was kept at 50 K which is higher than superconducting critical temperature ($T_c$). In addition, supplemental high-resolution ARPES measurement was performed at the beamline BL9A of HiSOR to check if the crossing of the LCB band can be seen by ARPES or not. The energy and angular resolution were set to 10 meV and < 0.004 Å$^{-1}$. Samples are cleaved in situ in the UHV chamber at room temperature and measured at 50 K.

**Data availability**. The data sets generated during and/or analyzed during the current study are available from the corresponding authors on reasonable request.

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

## Acknowledgements

The experiments were performed with the approval of Proposal Assessing Committee of the Hiroshima Synchrotron Radiation Center (Proposal No. 16AG052 and 16AU004). S. W., T.O. thank Professor Hitoshi Sato, Mr. Rousuli Awabaikeli, Mr. Ming-Tian Zheng, Dr. Eike Fabian Schwier and Professor Kenya Shimada for their support during

supplumental experiments at beamlines BL7 and BL1 in HiSOR. T.O. appreciates Professor Chikako Moriyoshi for her help in the preparation of experiment.

## Author contributions

S.W., K.S., K.T., T.Y, K.M, M.A., and T.O. carried out ARPES and SARPES measurement. M.N., S.W., and I.T. synthesized and characterized the single crystals. S.W. analyzed (S)ARPES data and wrote the manuscript with input from M.K.,Y.U., A.K., and T. O. T.O. conceived the experiment. All authors contributed to the scientific discussions.

## Additional information

**Competing interests:** The authors declare no competing financial interests.

