## [Peer Review File · Nature Communications]

Reviewers' comments:

Reviewer #1 (Remarks to the Author):

The paper of Wu et al. presents new spin-polarized photoemission data on an interesting superconductor $\text{LaO}_{0.55}\text{F}_{0.45}\text{BiS}_2$. The paper is well written and in my opinion presented conclusions are valid. Unfortunately, the amount of new physics is not sufficient for publication in Nature Comm.

The idea of the paper is twofold. Firstly, it shows the hidden spin polarization in the centrosymmetric crystal. Centrosymmetric crystal as a whole cannot exhibit spin-polarization, however, in certain cases the subsequent atomic layers in the real space may exhibit spin-momentum-locked spin polarization. This can be probed by angle-resolved photoemission (ARPES) because of its depth and momentum resolution.

Secondly the paper is motivated by the interesting predicted non-s-wave superconductivity in $\text{LaO}_{0.55}\text{F}_{0.45}\text{BiS}_2$, where spin-momentum locked Fermi surfaces are predicted to play crucial role.

New results in the current paper are experimental only. The theory has been published for example in the Refs. 18 (Yang et al.) and 19 (Dai et al.). High resolution ARPES data from $\text{LaO}_{0.55}\text{F}_{0.45}\text{BiS}_2$ has been published in Terashima et al. PRB 90, 220512 (2014) (Ref. 28). Therefore, as far as I understand, the novelty of the current manuscript is limited to the spin-polarized data from important features near the Fermi level. This is surely a very challenging experiment, however, the outcome is rather predictable, since a clear splitting has already been observed by Terashima et al., and the spin polarization modeled by the theory in Liu et al., PRB 91, 235204 (2015) (Ref. 10).

For these reasons I believe that the paper does not match the criteria of Nature Comm., however, certainly deserves to be published in one of the regular journals.

Independent on my decision I have three comments, which may help improving the manuscript:

1. The hidden spin polarization described in Zhang et al. Nature Phys. 10, 387 (2014) (Ref. 1) is compensated in adjacent layers. This means that in order to observe the spin-polarized photocurrent in ARPES the orientation of the surface layer must be the same over the macroscopic region of the beamspot. I am not sure about the details, but can imagine possible issues with terrace size and with the twinning domains. Authors should comment on that, and

possibly add some microscopic characterization of the surface (such as AFM micrograph). Within this context another publication on hidden spin polarization might be useful: Scientific Reports 6, 26197 (2016).

2. The spin-polarization in the ensemble of photoemitted electrons does not need to directly reflect the initial spin polarization (unless the spin-split spin-momentum-locked bands are 100% spin-polarized). There might be light-polarization dependence, and number of other matrix element effects. I would comment on that.

3. Liu et al., PRB 91, 235204 (2015) show pronounced surface-related features in band structure of LaOBiS₂. I believe possible implications of these were not discussed in text in detail.

Furthermore, although the manuscript is carefully written, there are still several formulations where the English should be improved.

Reviewer #2 (Remarks to the Author):

The hidden spin texture in LaOBiS₂ is an important topic and has been predicted. This manuscript presents a three-dimensional spin analysis of F-doped LaOBiS₂ and reveals the hidden spin texture. The main conclusion is there is a transition from Dresselhaus-like D2 to Rashba-Like R2 spin texture with varying energy. This paper presents a major progress in the spin physics of LaOBiS₂ family and I would recommend publication after addressing the technical issues below:

(1) There are already a few experimental realization of R-2 and D-2 examples in transition metal dichalcogenides. Since those are strongly related to this manuscript, they should be properly referenced.

(2) Although the spin-ARPES does suggest the band splitting of the valence band, it is useful to show the original data of Fig.1d and Fig.1e

Reviewer #3 (Remarks to the Author):

This manuscript details an ARPES and spin-resolved ARPES study of a LaOBiS₂ superconductor. While the bulk material is centrosymmetric, the local broken symmetry within the unit cell opens up avenues for both Rashba and Dresselhaus type spin-splitting of the valence and conduction bands. This type of spin-splitting is reversed from one layer to the other, ensuring zero spin polarization when averaging over the full unit cell, but has been a topic of recent interest and could offer potential applications. The added superconductivity present in the material adds high interest in this type of study due to enabling the exploration of the interplay of the spin-orbit interaction with superconductivity.

The present work includes ARPES data that shows band dispersions and constant-energy contours that are claimed to be consistent with expectations. While the data is not very clear on its own and needs to be displayed as second-derivative maps, it seems correctly interpreted. The greater impact is the spin-resolved ARPES data, which provides direct evidence of observable spin splitting of the surface electronic structure. The data is interpreted to demonstrate the predicted spin splitting of bulk states, and reasons that the data is consistent with a “R2” Rashba-type spin texture. The work also claims data consistent with an additional component of a “D2” Dresselhaus spin texture.

A direct spin-resolved ARPES investigation of an interesting material such as this will bring significant interest and have good impact in the field. The data appears well measured and interpreted in many places. The thorough investigation and analysis of the measured spin-texture gives high confidence in the identification of a Rashba-type spin texture. Due to the likely interest in this primary observation in the interesting field, I tend to feel it could be published in Nature Communications. I feel the current manuscript pushes some of the claims/interpretations a bit further than the data or current explanations warrant, and I would recommend a couple items be addressed before publication.

1) The spin-resolved data thoroughly shows a Rashba-type spin-texture of the surface electronic structure of the conduction band, which is the primary focus of the work. It is not clear to me how to solidly distinguish the observed texture as due to an “R1” or an “R2” effect. The authors note an earlier work (Ref. 28) that performs ARPES on the material at both 70 and 880 eV (more bulk-sensitive) photon energy, which shows a roughly similar (but difficult to resolve) Fermi surfaces, and is used as evidence that the electronic states measured with 70 eV photons are bulk states. However, this reference includes only a very rough Fermi surface measured at 880 eV, and does not show an actual dispersion cuts, and certainly does not show a splitting in the conduction band. Therefore, it doesn't seem clear to me that the present data can distinguish whether the observed spin texture is present in each layer throughout the bulk (an “R2” effect) versus just present at the broken symmetry at the surface (an “R1” effect). Indeed, the supplemental material for Ref. 28 states, “our core level spectra imply that the electronic states of the topmost BiS2 layer may not be exactly the same as those of bulk after cleaving.” While I agree that the data is consistent with the theoretically predicted “R2” effect in the bulk electronic structure, it does not seem correct to present the current data as able to experimentally distinguish it as “R2” versus “R1”.

2) While the majority of the experimental data focuses on the Rashba-type spin texture, parts of the text of the manuscript and abstract strongly push the concept of a Dresselhaus-type texture present, as well. This is a very interesting feature to investigate. However, the actual experimental evidence seems less compelling. While the manuscript includes 10 spin-resolved EDCs dedicated to thoroughly studying the Rashba-type spin texture through a range of

momenta and geometries, the manuscript includes only 2 EDCs meant as evidence of the Dresselhaus texture. This seems significantly less compelling. To claim this as a solid direct observation of the predicted Dresselhaus component, more data with more clarity should be presented. For instance, in Fig. 3, two EDC's, both at negative values of k_y are shown. While it seems that the spin texture reverses, consistent with predictions, the data is far from clear, with very small polarizations ($\sim < 5\%$ with lots of noise if I am reading the plots correctly). While I trust that the two data sets are correctly located both on the same side of $k_y=0$, there is no included discussion to directly address the fact that if EDC "b" were actually located at positive k_y , the data would simply be consistent with the Rashba component. More directly, the current interpretation would be significantly bolstered with additional spin-resolved EDCs at positive k_y the show similar spin reversals.

I believe these two points should be addressed for publication.

Reply to the referee #1

Reviewers' comments:

Reviewer #1 (Remarks to the Author):

The paper of Wu et al. presents new spin-polarized photoemission data on an interesting superconductor $\text{LaO}_{0.55}\text{F}_{0.45}\text{BiS}_2$. The paper is well written and in my opinion presented conclusions are valid. Unfortunately, the amount of new physics is not sufficient for publication in Nature Comm.

First of all, we appreciate very much the referee that he or she admitted that the sample that we have measured is interesting and our conclusions are valid.

Furthermore, we would acknowledge the referee noticing us the weak point of our manuscript. That is, we did not correctly describe the novelty of our study.

The idea of the paper is twofold. Firstly, it shows the hidden spin polarization in the centrosymmetric crystal. Centrosymmetric crystal as a whole cannot exhibit spin-polarization, however, in certain cases the subsequent atomic layers in the real space may exhibit spin-momentum-locked spin polarization. This can be probed by angle-resolved photoemission (ARPES) because of its depth and momentum resolution.

Secondly the paper is motivated by the interesting predicted non-s-wave superconductivity in $\text{LaO}_{0.55}\text{F}_{0.45}\text{BiS}_2$, where spin-momentum locked Fermi surfaces are predicted to play crucial role.

As the referee pointed out the important points of our manuscript is following. (1) We could observe the hidden spin polarization in centrosymmetric crystal by the help of surface sensitivity of the photoemission process. (2) Our results suggest that the system is important since one can investigate the role of spin-momentum locking in the mechanism of superconductivity of the system.

However, the one more important finding of our study is that we have observed the coexistence of Rashba and Dresselhaus effects in one material for the first time. This transition of spin texture from Dresselhaus-like to Rashba-like in the conduction band as a function of energy has not been explicitly discussed in the preceding theoretical papers (Ref. 1 and Ref 8) though the crossing of conduction band by the transition can be seen in the figure of the formally published papers. Namely this transition has been overlooked in the theoretical papers. To our knowledge this is the first observation of the coexistence of Rashba-like and Dresselhaus-like spin texture and we think the complex spin texture can also affect to the property or mechanism of the super conductivity of the materials. Therefore, we believe that our finding is very interesting and important and the topic definitely meets the criteria of Nature Communications.

We would apologize to the referee that in this sense our manuscript was not well written and we should have emphasized the point more apparently. Thus, in the revised manuscript we tried to make the point clearer and added some more sentences on this point.
(See that part of blue letters in page 13.)

New results in the current paper are experimental only. The theory has been published for example in the Refs. 18 (Yang et al.) and 19 (Dai et al.). High resolution ARPES data from $\text{LaO}_{0.55}\text{F}_{0.45}\text{BiS}_2$ has been published in Terashima et al. PRB 90, 220512 (2014) (Ref. 28). Therefore, as far as I understand, the novelty of the current manuscript is limited to the spin-polarized data from important

features near the Fermi level. This is surely a very challenging experiment, however, the outcome is rather predictable, since a clear splitting has already been observed by Terashima *et al.*, and the spin polarization modeled by the theory in Liu *et al.*, PRB 91, 235204 (2015) (Ref. 10).

For these reasons I believe that the paper does not match the criteria of Nature Comm., however, certainly deserves to be published in one of the regular journals.

We would once again apologize to the referee that we did not properly express the novelty of our study in the previous manuscript. As we already mentioned that the theoretical paper by Zhang *et al.* predicted the possible hidden Dresselhaus spin texture in the conduction band of the material but has not mentioned explicitly the change of spin texture from Dresselhaus-like to Rashba-like as a function of binding energy in the conduction band. We believe that our experimental finding of the transition from Dresselhaus-like spin texture to Rashba-like one in the conduction band is novel and opens new pathway to the new spintronics device as well as the further understanding of superconductivity.

In addition, as the referee pointed out the spin- and angle-resolved photoemission spectroscopy (spin-ARPES) is very challenging experiment and the information which one can obtain is much richer than normal ARPES. Namely, although the band splitting in the conduction band along X- Γ -X line has been demonstrated in the previous paper published by Terashima *et al.*, any information of spin along M-X-M line could not be revealed since the splitting along M-X-M line is too small to resolved by normal ARPES. Hence, no other experimental techniques than the spin-ARPES can address this important issue.

Therefore, even the ARPES study has been already published by Terashima *et al.* we are sure that the impact of our study is still substantial.

Independent on my decision I have three comments, which may help improving the manuscript:

1. The hidden spin polarization described in Zhang et al. Nature Phys. 10, 387 (2014) (Ref. 1) is compensated in adjacent layers. This means that in order to observe the spin-polarized photocurrent in ARPES the orientation of the surface layer must be the same over the macroscopic region of the beamspot. I am not sure about the details, but can imagine possible issues with terrace size and with the twinning domains. Authors should comment on that, and possibly add some microscopic characterization of the surface (such as AFM micrograph). Within this context another publication on hidden spin polarization might be useful: Scientific Reports 6, 26197 (2016).

We would thank the referee very much for the important comments. In case of $2H$ -WSe₂ or $2H$ -MoS₂ the unit cell of the sample consists of two trigonal structures oriented opposite direction and making hexagonal symmetry in total. Since the trigonal layers are bound by van der Waals force the sample can be cleaved at any van der Waals gap. Thus, the possibility of existence of domains possessing positive spin polarization and negative polarization is the same. Therefore, if the beam spot size is bigger than the domain size the spin polarization can be cancelled out in the observation. However, in case of LaO(F)BiS₂ in the unit cell the bond between BiS₂ layer and La₂O₂ layer is strong and the adjacent BiS₂ layers are bound by weak van der Waals force. Therefore, the sample is cleaved always at this van der Waals gap and there is only one type of termination and one does not have to consider the effect of domains possessing different spin polarization. We agree with referee that we did not mention this important point and in the revised manuscript we have added some explanation on this point. (See blue part in the text of page 5 and 12.)

2. The spin-polarization in the ensemble of photoemitted electrons does not need to directly reflect the initial spin polarization (unless

the spin-split spin-momentum-locked bands are 100% spin-polarized). There might be light-polarization dependence, and number of other matrix element effects. I would comment on that.

Thank you again for this important comment. We agree that the observed spin polarization by the spin-ARPES measurement can be affected by experimental geometry and so on relating the matrix element effect especially in the materials possessing strong spin-orbit interaction like topological insulators. Therefore, as discussed in page 9 we have measured the spin texture of conduction band at different X points (alpha and beta in Fig. 2a) where we need to perform spin-ARPES measurement with different experimental geometry and confirmed that the spin polarizations observed with different experimental geometries show the same spin texture. The absence of X and Z spin component in Fig. 2f is also implying that the observed spin polarization is not modified very much by the photoemission matrix element and so on.

3. Liu et al., PRB 91, 235204 (2015) show pronounced surface-related features in band structure of LaOBiS₂. I believe possible implications of these were not discussed in text in detail.

We agree with the referee that we have not discussed very much about the validity that the observed spin polarization is due to hidden bulk Rashba or Dresselhaus effect instead of surface Rashba effect. In the previous manuscript we argued that the resemblance of Fermi surface taken with $h\nu=18$ eV and 70 eV with the one observed with bulk sensitive soft X-ray (880 eV) and the consistent results with theoretical calculation for the bulk state are the collateral evidences that our observed spin polarization is due to bulk contribution. However, we have noticed that the fact that we have observed the transition of Rashba and Dresselhaus-like spin texture itself is the strongest evidence that the observed spin texture is not due to simple Rashba effect but due to bulk electronic structure. To make this point

clearer we added this argue in the revised manuscript. (Blue part from the end of page 11 to 12.)

Furthermore, although the manuscript is carefully written, there are still several formulations where the English should be improved.

We checked once again the manuscript and corrected some English expressions.

Reply to the referee #2

Reviewer #2 (Remarks to the Author):

The hidden spin texture in LaOBiS₂ is an important topic and has been predicted. This manuscript presents a three-dimensional spin analysis of F-doped LaOBiS₂ and reveals the hidden spin texture. The main conclusion is there is a transition from Dresselhaus-like D₂ to Rashba-Like R₂ spin texture with varying energy. This paper presents a major progress in the spin physics of LaOBiS₂ family and I would recommend publication after addressing the technical issues below:

First of all, we would greatly appreciate the positive comments by the referee.

(1) There are already a few experimental realization of R-2 and D-2 examples in transition metal dichalcogenides. Since those are strongly related to this manuscript, they should be properly referenced.

Thank you very much for the important comments. We checked previous papers and referred following papers in the revised manuscript.

Gehlmann, M. et al. Quasi 2D electronic states with high spin-polarization in centrosymmetric MoS₂ bulk crystals. *Sci. Rep.* **6**, 26197 (2016).

Yao, W. et al. Direct observation of spin-layer locking by local Rashba effect in monolayer semiconducting PtSe₂ film. *Nat Commun.* **8**, 14216 (2017).

(2) Although the spin-ARPES does suggest the band splitting of the valence band, it is useful to show the original data of Fig.1d and Fig.1e

In accordance with the referee's comment we presented the original data of Fig. 1d and 1e as well as Fig. 2c in Figure S1 of Supplementary information.

Reply to the referee #3

Reviewer #3 (Remarks to the Author):

This manuscript details an ARPES and spin-resolved ARPES study of a LaOBiS₂ superconductor. While the bulk material is centrosymmetric, the local broken symmetry within the unit cell opens up avenues for both Rashba and Dresselhaus type spin-splitting of the valence and conduction bands. This type of spin-splitting is reversed from one layer to the other, ensuring zero spin polarization when averaging over the full unit cell, but has been a topic of recent interest and could offer potential applications. The added superconductivity present in the material adds high interest in this type of study due to enabling the exploration of the interplay of the spin-orbit interaction with superconductivity.

The present work includes ARPES data that shows band dispersions and constant-energy contours that are claimed to be consistent with expectations. While the data is not very clear on its own and needs to be displayed as second-derivative maps, it seems correctly interpreted. The greater impact is the spin-resolved ARPES data, which provides direct evidence of observable spin splitting of the surface electronic structure. The data is interpreted to demonstrate the predicted spin splitting of bulk states, and reasons that the data is consistent with a “R2” Rashba-type spin texture. The work also claims data consistent with an additional component of a “D2” Dresselhaus spin texture.

A direct spin-resolved ARPES investigation of an interesting material such as this will bring significant interest and have good impact in the field. The data appears well measured and interpreted in many places. The thorough investigation and analysis of the measured

spin-texture gives high confidence in the identification of a Rashba-type spin texture. Due to the likely interest in this primary observation in the interesting field, I tend to feel it could be published in Nature Communications. I feel the current manuscript pushes some of the claims/interpretations a bit further than the data or current explanations warrant, and I would recommend a couple items be addressed before publication.

First of all, we would appreciate the referee very much for the careful reading of our manuscript the positive comments on it.

1) The spin-resolved data thoroughly shows a Rashba-type spin-texture of the surface electronic structure of the conduction band, which is the primary focus of the work. It is not clear to me how to solidly distinguish the observed texture as due to an “R1” or an “R2” effect. The authors note an earlier work (Ref. 28) that performs ARPES on the material at both 70 and 880 eV (more bulk-sensitive) photon energy, which shows a roughly similar (but difficult to resolve) Fermi surfaces, and is used as evidence that the electronic states measured with 70 eV photons are bulk states. However, this reference includes only a very rough Fermi surface measured at 880 eV, and does not show an actual dispersion cuts, and certainly does not show a splitting in the conduction band. Therefore, it doesn't seem clear to me that the present data can distinguish whether the observed spin texture is present in each layer throughout the bulk (an “R2” effect) versus just present at the broken symmetry at the surface (an “R1” effect). Indeed, the supplemental material for Ref. 28 states, “our core level spectra imply that the electronic states of the topmost BiS2 layer may not be exactly the same as those of bulk after cleaving.” While I agree that the data is consistent with the theoretically predicted “R2” effect in the bulk electronic structure, it does not seem correct to present the

current data as able to experimentally distinguish it as “R2” versus “R1”.

Thank you very much for this important comment.

We agree with referee that our discussion by just comparing bulk sensitive ARPES results and our results are rather weak. However, we think that the good agreement of observed band structure (energy gap size, size of splitting, Rashba parameter and so on) and band calculation for the bulk system fairly suggests that our observed spin texture is due to bulk electronic states. Furthermore, we would like to emphasize that the observed transition from Dresselhaus-like to Rashba-like spin texture is one of the direct evidence that the observed spin structure is not by the trivial surface Rashba state by the breaking of inversion symmetry. Thus, in the revised manuscript we added this discussion as in the blue part from the end of page 11 to page 12 to make the manuscript more convincing.

2) While the majority of the experimental data focuses on the Rashba-type spin texture, parts of the text of the manuscript and abstract strongly push the concept of a Dresselhaus-type texture present, as well. This is a very interesting feature to investigate. However, the actual experimental evidence seems less compelling. While the manuscript includes 10 spin-resolved EDCs dedicated to thoroughly studying the Rashba-type spin texture through a range of momenta and geometries, the manuscript includes only 2 EDCs meant as evidence of the Dresselhaus texture. This seems significantly less compelling. To claim this as a solid direct observation of the predicted Dresselhaus component, more data with more clarity should be presented. For instance, in Fig. 3, two EDC's, both at negative values of k_y are shown. While it seems that the spin

texture reverses, consistent with predictions, the data is far from clear, with very small polarizations ($\sim < 5\%$ with lots of noise if I am reading the plots correctly). While I trust that the two data sets are correctly located both on the same side of $k_y=0$, there is no included discussion to directly address the fact that if EDC “b” were actually located at positive k_y , the data would simply be consistent with the Rashba component. More directly, the current interpretation would be significantly bolstered with additional spin-resolved EDCs at positive k_y the show similar spin reversals.

We would thank again to the referee for this comment that can improve our manuscript. We fully agree with the referee it is better if we can show more spin-ARPES data at the other side of the band along M-X-M line. Although the signal to noise ratio are not good we present the data set which covers from negative to positive k_y values with respect to X point in supplementary materials. (See Figure S5)

Unfortunately, we do not have the data very near to X point in positive k_y value. However, we can see the clear evidence of peak shifting and the reversal of spin polarization at the region of Rashba-like spin texture (1 and 7 & 8 in Figure S5b) and also the reduction of spin polarization and the tendency of spin reversal toward X point both in the negative k_y region (from 1 to 4) and positive k_y region (from 8 to 6). We believe that these additional data are enough to convince the readers that the system shows transition of spin texture from Rashba-like to Dresselhaus-like and the observed spin texture is due to bulk spin-electronic states.

Reviewers' comments:

Reviewer #1 (Remarks to the Author):

I have no comments for Authors at this time.

Reviewer #2 (Remarks to the Author):

The authors have addressed my questions and I would recommend publication.

Reviewer #3 (Remarks to the Author):

The authors have submitted a slightly revised manuscript that moves towards addressing some of the referees' comments, and have made revisions to address my comments. In particular, additional data was added to the Supplementary Information to address my second comment. While this change is an improvement, focusing on this new data has brought a few points to my attention.

1) In figure 2a, "Cut 3" is shown to cut through the X point along k_x at a fixed k_y . As Figure 3a is an ARPES cut along "Cut3" the horizontal axis should read " k_x ", however the manuscript has it read " k_y ". This inconsistency is repeated in figure S5, as well as the author's text in their reply to my comment. (I note that in my original comment, I also used " k_y " when I believe " k_x " was correct, but was because of the error in the figure axis labeling I did not notice before.) I assume this is just a mistake, but the repeated use of this needs to be addressed and confirmed.

2) In my original second comment, I asked for more spin-resolved EDCs along MXM to more solidly confirm the presence of the Dresselhaus-type texture (which now seems to be the main point of the manuscript). This has been addressed with the additional data in the supplement, which although is not as clear as one might like, it is at least consistent with the interpretation as described by the authors. I would like to also note that in the authors efforts to convince of the Rashba-type texture, data was shown along "cut1" and "cut2". As I understand the diagrams, "cut3" should be identical to "cut2", and so "cut2" should also demonstrate the energy dependant switch from Rashba to Dresselhaus textures. Can data be shown, and if not, why is this effect not seen along "cut2" and only along "cut3"? If the effect is for some reason geometry dependent, it may cast doubt on the interpretation, and should be addressed.

3) Again, in more careful inspection of the new data, I notice that in presenting the polarization

curves of the spin-polarized EDCs, the authors tend to plot the data using the technique of “filling-to-zero” of filling in positive values with red coloring down to the horizontal axis, and filling in negative values blue up to the horizontal axis. In particular with noisy data, this helps inspect the data in a kind of binary “up/down” way. While I understand the method, I did not before notice that this plotting approach was done “selectively” – in many cases only the desired parts of the data were filled in, while other areas of the data are not “filled-in” even when the data value is not zero. I also understand that this approach may be used to act as a “guide-to-the-eye” to focus attention on the aspects of the data the authors want the reader to see. However, I feel this is quite misleading and misrepresents the actual data which should be presented in a consistent manner globally. If the “filled-in” technique is desired, it must be used through the entire EDC so the reader can correctly view the real data accurately.

While the manuscript has been improved, I believe these issues should be addressed before publication.

Reply to the referee #1

I have no comments for Authors at this time.

Thank you very much for the referee's careful reading of our manuscript. We appreciate very much that the referee is satisfied with our answers and understood the importance of our results.

Reply to the referee #2

The authors have addressed my questions and I would recommend publication.

We are grateful for the referee's careful reading of our manuscript and appreciate very much for the referee's positive comment to publish our results.

Reply to the referee #3

The authors have submitted a slightly revised manuscript that moves towards addressing some of the referees' comments, and have made revisions to address my comments. In particular, additional data was added to the Supplementary Information to address my second comment. While this change is an improvement, focusing on this new data has brought a few points to my attention.

First of all, we thank to the referee very much for his/her careful reading of our manuscript and important comments that help to improve our manuscript.

We would thank the referee once again that the comments or suggestions

from the referee were really helpful to improve our manuscript or correct some mistakes in our previous manuscript.

We would like to answer to the comments as follows.

Reviewers' comments:

- 1) In figure 2a, "Cut 3" is shown to cut through the X point along k_x at a fixed k_y . As Figure 3a is an ARPES cut along "Cut3" the horizontal axis should read " k_x ", however the manuscript has it read " k_y ". This inconsistency is repeated in figure S5, as well as the author's text in their reply to my comment. (I note that in my original comment, I also used " k_y " when I believe " k_x " was correct, but was because of the error in the figure axis labeling I did not notice before.) I assume this is just a mistake, but the repeated use of this needs to be addressed and confirmed.

We would like to apologize for the mistake and would like to thank to the referee for this important indication.

As the referee commented this is just a mistake and we corrected them in the revised manuscript.

- 2) In my original second comment, I asked for more spin-resolved EDCs along MXM to more solidly confirm the presence of the Dresselhaus-type texture (which now seems to be the main point of the manuscript). This has been addressed with the additional data in the supplement, which although is not as clear as one might like, it is at least consistent with the interpretation as described by the authors. I would like to also note that in the authors efforts to convince of the Rashba-type texture, data was shown along "cut1" and "cut2". As I understand the diagrams, "cut3" should be identical to "cut2", and so "cut2" should also demonstrate the energy dependant switch from Rashba to

Dresselhaus textures. Can data be shown, and if not, why is this effect not seen along “cut2” and only along “cut3”? If the effect is for some reason geometry dependent, it may cast doubt on the interpretation, and should be addressed.

We thank to the referee that he or she understood the main point of our manuscript.

We would also appreciate that the referee agrees that the data of cut3 is consistent with our interpretation.

As for the cut2, unfortunately we did not measure the spin-EDCs at the k-points near the X point since we were concentrating to measure the spin polarization of the state and chose the k-points where are well apart from the X point since we thought we can measure the spin-polarization easily. Unfortunately, at that time we had not yet noticed the switch of the spin texture from Rashba type to Dresselhaus type in the state and did not measure the spin polarization near the X point.

So we have the data of cut2 only at $|k| > 0.1 \text{ \AA}^{-1}$.

However, during measuring the cut3 we found the change of spin polarization from Rashba type to Dresselhaus type at some k-point where are in between X point and $|k| \leq 0.1 \text{ \AA}^{-1}$

Since the beam time of synchrotron radiation facility is limited and we did not have enough time to repeat the measurement near X point in cut2.

However, from the consistency of all the other data of cut2 and cut3 (spin-ARPES data at the k points where are $|k| > 0.1 \text{ \AA}^{-1}$, as well as the band dispersion obtained by normal ARPES and so on) we believe that the spin texture switch is universal for the band and not only at cut3.

To make this point clearer we added following text in the revised supplementary note 4.

Note that since the spin-EDC spectra of cut 2 are taken only at k-points at

$|k| > 0.1 \text{ \AA}^{-1}$ we did not see the spin polarization change from Rashba type to

Dresselhaus type (see Fig.2c and 2e of the main text).

3) Again, in more careful inspection of the new data, I notice that in presenting the polarization curves of the spin-polarized EDCs, the authors tend to plot the data using the technique of “filling-to-zero” of filling in positive values with red coloring down to the horizontal axis, and filling in negative values blue up to the horizontal axis. In particular, with noisy data, this helps inspect the data in a kind of binary “up/down” way. While I understand the method, I did not before notice that this plotting approach was done “selectively” – in many cases only the desired parts of the data were filled in, while other areas of the data are not “filled-in” even when the data value is not zero. I also understand that this approach may be used to act as a “guide-to-the-eye” to focus attention on the aspects of the data the authors want the reader to see. However, I feel this is quite misleading and misrepresents the actual data which should be presented in a consistent manner globally. If the “filled-in” technique is desired, it must be used through the entire EDC so the reader can correctly view the real data accurately.

We again appreciate the referee for his/her very careful reading of our manuscript.

Actually we used the “filling-to-zero” technique to show our spin-polarization data that we thought it acts as a “guide-to-the-eye” as the referee commented.

We thought the data is always especially noisy above Fermi level (because of the small number of counts) and did not use the technique above Fermi level. In addition, we also thought it might be easier for readers to understand the data we omitted some parts of the data to fill to zero.

However, we also agree to the referee that this kind of selection sometimes misleads readers. Since we believe that the interpretation of the data is consistent and unchanged even with using the “fill-to-zero” technique all the parts we decided to use the technique for all the parts in the revised figures.

While the manuscript has been improved, I believe these issues should be addressed before publication.

Thank you very much again for the positive comment. We believe that now we have addressed all the issues raised by the referee properly and the manuscript is ready to publication.

Reviewers' Comments:

Reviewer #3 (Remarks to the Author):

I am satisfied with the authors' replies and corrections, and appreciate their time and patience in addressing my concerns. I feel the manuscript is suitable for publication.

Reply to the referee #3

Reviewer #3 (Remarks to the Author):

I am satisfied with the authors' replies and corrections, and appreciate their time and patience in addressing my concerns. I feel the manuscript is suitable for publication.

Our answer to the referee #3

Thank you very much for the careful readings and comments and suggestions that improved our manuscript very much. We appreciate very much for the positive comments of the referee for publication.